**Data Availability Statement:** Raw sRNA data are available at the NCBI as the accession no.

# Identification and expression analysis of miRNAs and elucidation of their role in salt tolerance in rice varieties susceptible and tolerant to salinity

Shaifaly Parmar[1©], Sachin Ashruba Gharat[1©¤], Ravichandra Tagirasa[1], Tilak Chandra[1], Lambodar Behera[2], Sushant Kumar Dash[2], Birendra Prasad Shaw[1]*

**1** Abiotic Stress and Agro-Biotechnology Lab, Institute of Life Sciences, Bhubaneswar, Odisha, India, **2** Crop Improvement Division, ICAR-National Rice Research Institute (Formerly Central Rice Research Institute), Cuttack, Odisha, India

© These authors contributed equally to this work.
¤ Current address: Biochemical Sciences Division, CSIR-National Chemical Laboratory, Pune, India
* b_p_shaw@yahoo.com

## Abstract

Soil salinization is a serious problem for cultivation of rice, as among cereals rice is the most salt sensitive crop, and more than 40% of the total agricultural land amounting to approximately 80 million ha the world over is salt affected. Salinity affects a plant in a varieties of ways, including ion toxicity, osmotic stress and oxidative damage. Since miRNAs occupy the top place in biochemical events determining a trait, understanding their role in salt tolerance is highly desirable, which may allow introduction of the trait in the rice cultivars of choice through biotechnological interventions. High throughput sequencing of sRNAs in the root and shoot tissues of the seedlings of the control and NaCl treated Pokkali, a salt-tolerant rice variety, identified 75 conserved miRNAs and mapped 200 sRNAs to the rice genome as novel miRNAs. Expression of nine novel miRNAs and two conserved miRNAs were confirmed by Northern blotting. Several of both conserved and novel miRNAs that expressed differentially in root and/or shoot tissues targeted transcription factors like AP2/EREBP domain protein, ARF, NAC, MYB, NF-YA, HD-Zip III, TCP and SBP reported to be involved in salt tolerance or in abiotic stress tolerance in general. Most of the novel miRNAs expressed in the salt tolerant wild rice *Oryza coarctata*, suggesting conservation of miRNAs in taxonomically related species. One of the novel miRNAs, osa-miR12477, also targeted L-ascorbate oxidase (*LAO*), indicating build-up of oxidative stress in the plant upon salt treatment, which was confirmed by DAB staining. Thus, salt tolerance might involve miRNA-mediated regulation of 1) cellular abundance of the hormone signaling components like EREBP and ARF, 2) synthesis of abiotic stress related transcription factors, and 3) antioxidative component like LAO for mitigation of oxidative damage. The study clearly indicated importance of osa-miR12477 regulated expression of *LAO* in salt tolerance in the plant.

GSE133866 https://www.ncbi.nlm.nih.gov/geo/query/acc.cgi?acc=GSE133866.

**Funding:** BPS EMR/2016/001139 Science and Engineering Research Board, New Delhi http://www.serb.gov.in/home.php The funders had no role in study design, data collection and analysis, decision to publish, or preparation of the manuscript.

**Competing interests:** The authors have declared that no competing interests exist.

## Introduction

Rice, undisputedly, is one of the most important crops in the world, not only because of its importance as staple food, but also because of several other features like vast availability of germplasm, including landraces and wild progenitors varying in innumerable characters, a comparatively smaller genome size and ease of transformation and regeneration that makes it a model cereal system. It has been genetically modified successfully for nutritional quality and resistance to various diseases [1]. However, the major challenge lies in increasing its yield with respect to the ever-inflating population of the world, as the impact of several abiotic stresses negatively influences its yield [2].

Among the abiotic stresses, the magnitude of salinity problem in rice cultivation seems to be huge, as more than 80 million ha of agricultural land representing 40% of total irrigated land the world over are affected from salinization [3] and more than 50% of all arable lands are likely to suffer from soil salinization due to irrigation and climate change by 2050 [4]. Salinity is especially a threat to rice production because among cereals rice is the most salt-sensitive crop [5] and both seedling and reproductive stages of the plant are highly sensitive to salinity [6–9]. Thus, increasing salinization of cultivable land is in contrast to the demand for rice production in future, which should must increase from the current production of approximately 500 million tons to at least 800 million tons [10] to meet the demand of estimated 9.6 billion people by 2050 [11]. This requirement is in the backdrop of challenges being faced by the staple food crops from abiotic stresses, including salinity, that negatively influence their biomass production and yield [12]. Breeding programmes on development of rice cultivars tolerant to salinity along with maintaining the yield and quality to achieve the projected target of rice production has so far not been successful. Hence, the option left is to introduce the trait in the rice cultivar of choice through biotechnological intervention.

Salinity is most complex in action among the abiotic stresses, as it invokes both ionic and osmotic actions that increases its toxicity potential greatly. Plants have developed a variety of power of adaptation to abiotic stresses, including salinity stress, involving cascades of signaling networks, which ultimately lead to changes in expression of many genes determining the fate of cellular, biochemical and physiological processes important from the view of tolerance to the stress [13]. Many such genes encoding effector and regulator proteins have been identified that impart resistance to salinity to plants, including those involved in maintenance of ion homeostasis, osmotic protection through compatible solute accumulation, protection against oxidative stress, polyamine accumulation and transcriptional regulation [14]. Efficacy of a few of them in the salt tolerance process has also been demonstrated in model plants by generating transgenic lines overexpressing genes like betaine aldehyde dehydrogenase (*BADH*), high-affinity potassium transporter (*HKT*), $\Delta^1$-pyrroline-5-carboxylate synthase (*P5CS*), vacuolar $H^+$-pyrophosphorylase, etc. [14,15]. However, it has been increasingly realized that salt tolerance besides being a quantitative trait requires precise regulation of expression of the concerned genes, which may be upregulation of a few while downregulation of others. In this context microRNAs (miRNAs), the switch at the level of post-transcriptional gene regulation, as well as at the level of transcription [16,17] and translation [18], could be playing important role. Hence, identification of these miRNAs could provide a better strategy for engineering salt tolerance in plants than that accomplished by regulatory elements or effectors [19]. The view also stems from the fact that their expression is regulated by environmental cues, besides having multiple targets [20,21].

MicroRNAs are pervasive members of small RNA family, which smartly regulate the gene expression at the post-transcriptional and translational levels [22]. Unlike animals, plant miRNAs have very high or near-perfect complementarity with the target mRNA. The miRNA gene

is transcribed in a normal transcriptional fashion by RNA polymerase II into a primary miRNA, which is further cleaved by DCL1/HYL1 along with supporting molecules, into a precursor miRNA [23]. This precursor forms a characteristic stem loop structure bearing the mature miRNA and miRNA* sequence in opposite arms. In plants, this precursor miRNA gets methylated at 3' end by HEN1 and is brought to the cytoplasm by HST1. The miRNA-miRNA* duplex is cleaved and the resulting mature miRNA is loaded on to the RISC assembly by arganoute protein, while the miRNA* gets degraded [24]. A mature miRNA inhibits gene expression either directly by stalling the translation of proteins or indirectly by targeting the cognate mRNAs [25]. Post transcriptional regulation of genes has increasingly been realized to be the most conserved gene regulatory mechanism important for development, environmental stress responses and varieties of biological processes in eukaryotes [26,27].

The advent of next generation sequencing (NGS) has greatly enriched the database of miRNAs. For plants so far 10405 miRNAs sequences from 82 species have been deposited in miRBase 22.1 (http://www.mirbase.org). The occurrence of miRNAs in such a huge number, which is still increasing, is because of the fact that the profile of miRNAs expression differs greatly from species to species, both quantitatively and qualitatively. Besides, even within a species it is expected that the miRNA expression profile would differ from cultivar to cultivar depending on the trait with regard to which the two cultivars differ, and on the same logic it is also expected that their expression profile would differ considerably in the plant under contrasting environmental conditions [28,29]. Species-dependent response of miRNAs to salt treatment has been seen in many plants, including glycophytes and halophytes [21,29–39]. However, report on comparative study of expression profile of miRNAs in rice cultivars, or in any crop cultivars, contrast for salt tolerance, or tolerance to any abiotic stress is scant. Therefore, the current study was planned to see the expression profile of miRNAs in a salt-tolerant rice cultivar Pokkali in presence and absence of NaCl and to check the response pattern of a few important salt-responsive miRNAs in a salt-sensitive rice cultivar Badami in order to identify the miRNAs and their targets that could be involved in the salt tolerance process considering that the findings could be useful in improving salt tolerance in rice cultivar of interest through biotechnological interventions.

## Results

### Small RNA sequencing

More than 21 million high quality reads of sRNA were obtained from the four sRNA libraries (control root- CR, control shoot- CS, 256 mM NaCl (EC ~21 mS/cm) treated root- TR and 256 mM NaCl treated shoot- TS) sequenced (Table 1). These represented 99.72%, 99.78%, 99.86% and 99.87% of the total sequence reads from CR, CS, TR and TS, respectively. Total reads considered for finding the known miRNAs and predicting the novel miRNAs were 6887961, 13116913, 12126876 and 9765004 in CR, CS, TR and TS, respectively after removing the reads of length less than 16 nucleotides (nt) and more than 35 nt, low complexity reads, reads with T/R RNA matches and invalid sequence reads (Table 1). The non-redundant reads of length between 16-nt and 35-nt were 1143060, 1664461, 1219751 and 1035994, respectively for CR, CS, TR and TS. Most of the sRNA non-redundant sequences from all the libraries were 23-nt to 24-nt long (S1 File). Although the unique reads were more than one million in each of the libraries, and nearly 20% of the total unique reads, the total number of miRNAs discovered in CR, CS, TR and TS were only 147, 418, 261 and 357, respectively. Of these, 29, 35, 25 and 39 were conserved and 118, 383, 236 and 318 were mapped as novel miRNAs in CR, CS, TR and TS, respectively. The number of conserved miRNAs decreased, while that of the predicted (novel) miRNAs increased in root in response to the salt treatment (Table 1).

**Table 1. Summary of the sRNA reads.** Adapter ligated libraries of sRNAs from the root and shoot. tissues of *O. sativa* cv. Pokkali seedlings, control and treated with 256 mM NaCl for 9 h on the 9th day of germination were sequenced on Illumina platform. The sequences were processed with various bioinformatics tools to get the putative miRNAs.

| Read Distribution | Root control | Root treated | Leaf control | Leaf treated |
|---|---|---|---|---|
| Total no. of raw reads | 21291582 | 22498709 | 24465280 | 22021240 |
| Total no. of HQ filtered reads after first QC | 21232204 | 22467178 | 24412174 | 21993648 |
| Total no of reads after adapter trimming and length filtration (Min length = 15) | 19549175 | 21090549 | 24182661 | 21342069 |
| Total no of adapter trimmed, length filtered HQ reads after second QC | 19548882 | 21090188 | 24182301 | 21341350 |
| Reads unaligned with NONCODE | 18236576 | 20103872 | 22753639 | 18112677 |
| Filtered reads with (length (<16->35 nt), low complexity, T/R RNA match, Invalid) which are discarded | 11348615 | 7976996 | 9636726 | 8347673 |
| Total Reads Used for Known and Novel miRNA Prediction | 6887961 | 12126876 | 13116913 | 9765004 |
| Known miRNA | 29 | 25 | 35 | 39 |
| Novel miRNA | 118 | 236 | 383 | 318 |
| Total no. of miRNA | 147 | 261 | 418 | 357 |

However, opposite result was obtained in shoot, i.e. the number of conserved miRNAs increased, while that of the novel miRNAs decreased in response to the salt treatment (Table 1).

## Family distribution and Northern blot analysis of the conserved miRNAs

Both root and shoot tissues showed great difference in composition of the conserved miRNAs under control and salt treated conditions (S2 File). The conserved miRNAs identified belonged to 35 families (S2 File, S3 File). Responses of two of the conserved miRNAs identified, namely miR159a and miR167f, to salt application were checked by Northern blot analysis. The result of the Northern blot confirmed that these two miRNAs expressed both in root and shoot tissues of not only the salt-tolerant Pokkali, but also of the non-tolerant Badami (Fig 1, S4 File). These miRNAs also showed noticeable changes in expression in response to the salt treatment. The expression of miR159a was downregulated in both root and shoot in the salt-tolerant Pokkali, but was downregulated only in root in the salt-sensitive Badami in response to the salt application; in Badami shoot, miR159a showed upregulation in response to salt treatment. In contrast to miR159a, miR167f showed considerable downregulation only in the root of Pokkali, and its expression increased in both root and shoot of Badami.

## Bioinformatics of the novel miRNAs

Out of hundreds of novel miRNAs predicted by Illumina sequencing followed by bioinformatics analysis (Table 1), expression of at least nine of them could be successfully validated in Pokkali, as well as in Badami by Northern. The details of their bioinformatics qualifying parameters, like A+U percentage, minimum free energy (MFE) of the precursor and length of the precursor (PL, precursor length) are given in Table 2 and S5 File. The MFE values of the precursors varied from -34.1 kcal/mol to 67.7 kcal/mol, while the MFE index (MFEI) of the precursors varied from 0.67 to 1.72. A+U content in the precursors varied from 37.0% to 71.4%. The stem-loop structures obtained for the individual precursors by mFold software revealed the presence of the predicted individual miRNAs sequences in one of their arm (S5 File). The length of the novel miRNAs varied from 20-nt to 24-nt, and they varied greatly in abundance, expressed in reads per million (RPM). For several of them the miRNA* sequences were found to be present with miRNA/miRNA* mismatches varying from 3–5 (Table 2).

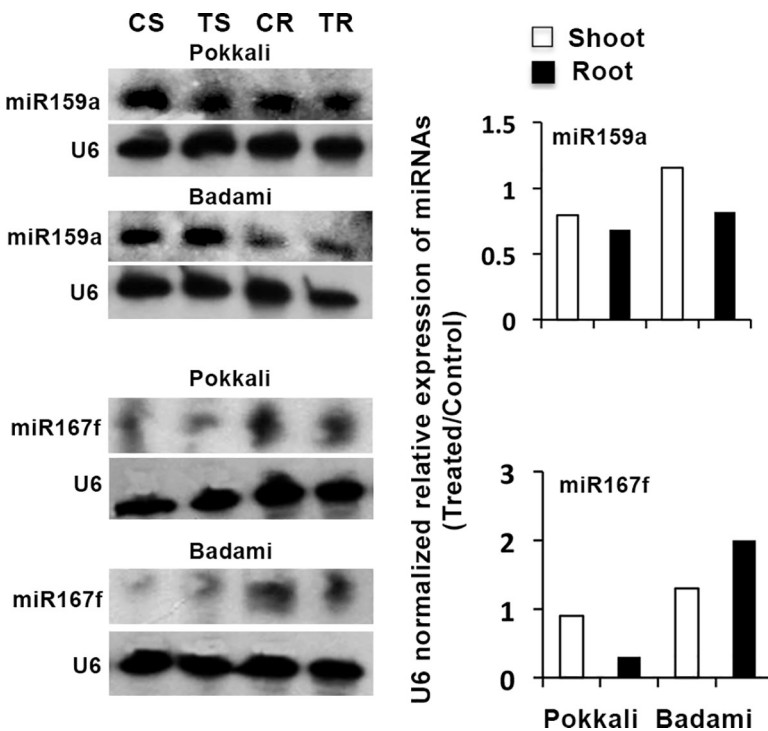

**Fig 1. Northern blots showing expression of two conserved miRNAs in root (R) and shoot (S) tissues of control (C) and 256 mM NaCl treated (T) seedlings of *O. sativa* cv. Pokkali (salt-tolerant) and *O. sativa* cv. Badami (salt-sensitive) on 9th day of germination.** The right panel shows the relative expression of the individual miRNAs in the root/shoot tissues of the treated seedlings over control ones, as obtained by densitometric analysis of the band intensity considering the band intensity of U6 for normalization.

### Northern blot analysis of the novel miRNAs

The results of the Northern for the novel miRNAs revealed great variation in their tissue-specificity as well as salt responsiveness in expression (Fig 2, S4 File). Among the novel miRNAs found expressing in the seedlings, osa-miR12475, osa-miR12476 and osa-miR12479 expressed only in root (Fig 2). Expression of the other miRNAs, like osa-miR12474, osa-miR12477, osa-miR12478, osa-miR12480, osa-miR12481 and osa-miR12482 was found both in root and shoot. Among the miRNAs that expressed only in root, the expression of osa-miR12475 and osa-miR12479 decreased both in Pokkali and Badami in response to NaCl. The decrease in expression of osa-miR12479 in response to the salt treatment was, however, much more in Badami compared with that in Pokkali. In oppose to osa-miR12475 and osa-miR12479, the expression of osa-miR12476 increased in response to the salt treatment in both Pokkali and Badami. Among the novel miRNAs showing expression in both root and shoot, the expression of osa-miR12478 was more prominent in root than in shoot, while the expression of osa-miR12477 was more prominent in shoot than in root in both Pokkali and Badami. The expression of osa-miR12478, however, decreased in shoot in both Pokkali and Badami in response to the NaCl application, in contrast to osa-miR12477, which showed NaCl-induced upregulation in expression in both root and shoot of both the varieties. In contrast to osa-miR12477 and osa-miR12478, osa-miR12474 showed differential expression pattern, particularly in shoot where its expression was upregulated in Badami, but downregulated in Pokkali in response to the salt treatment. In root the expression of osa-miR12474 decreased both in Pokkali and Badami in response to the salt treatment, comparatively more in the latter than in the former.

**Table 2. The novel miRNAs predicted by alignment of the sRNA sequences with rice genome database and confirmed for their expression by Northern in root and shoot tissues of the seedlings of *O. sativa* cv. Pokkali along with the relevant values/information of the basic bioinformatics criteria qualifying them as miRNAs.** MFE- Minimum Free Energy, MFEI- MEF index.

| Novel miRNAs | Sequence | Maximum abundance in tissue in reads per million (RPM) | Length (nt) | miRNA* | miRNA/ miRNA* mis-matches (nt) | Pre- cursor Length (nt) | A+U (%) | MFE (Kcal/ mol) | MFEI |
|---|---|---|---|---|---|---|---|---|---|
| osa-miR12474 | GCCCCGCGTCGCACGGATTCGT | 2212.9 | 22 | CTGAATCCTTTGCAGACGACT | 4 | 127 | 37.0 | -57.33 | 0.71 |
| osa-miR12475 | ACCGAGGCGCGTCAATTGCTG | 27.54 | 21 | AATGACGCAGCTTATGAGGTT | 4 | 175 | 46.3 | -67.7 | 0.72 |
| osa-miR12476 | ATTAATAGGGACAGTCGGGGGC | 10.52 | 22 | NO | | 114 | 61.4 | -34.9 | 0.81 |
| osa-miR12478 | CGGGGATGGAGCGACAGAAGCA | 56.22 | 22 | NO | | 100 | 51.2 | -34.1 | 0.67 |
| osa-miR12477 | TTGAGTGCAGCGTTGATGAACC | 24.33 | 22 | TTCACCAGCACTGCACCCAATC | 3 | 129 | 38.0 | -56.7 | 0.90 |
| osa-miR12479 | TTAGTTCACATCAATCTTCCT | 8.27 | 21 | NO | | 99 | 55.6 | -58.9 | 1.34 |
| osa-miR12480 | CAGCCCCGCGTCGCACGGATTCGT | 15.10 | 24 | GGCTGAATCCTTTGCAGACGACTT | 6 | 127 | 37.0 | -56.33 | 0.71 |
| osa-miR12481 | TATTATAAGACGTTTTGACT | 7.98 | 20 | NO | | 115 | 71.3 | -53.6 | 1.62 |
| osa-miR12482 | AGCAAGATATTGGGTATTTCTTTT | 0.72 | 24 | CTAGAAATACCCAATATCTTGCTG | 3 | 70 | 71.4 | -34.4 | 1.72 |

Unlike osa-miR12474, osa-miR12478 showed upregulation of expression in root in Pokkali and downregulation of expression in shoot in Badami in response to the salt treatment. Thus, osa-miR12474 and osa-miR12478 showed opposite expression pattern in root and shoot of the two rice varieties in response to the salt treatment. The expression of osa-miR12481 showed a pattern of decrease in shoot and increase in root both in Pokkali and Badami in response to the salt application with shoot showing a greater decrease in Pokkali than that in Badami. The miRNAs osa-miR12480 and osa-miR12482 showed similar response to the salt application in Pokkali; their expression increased both in root and shoot. Their response to salt application was different for root and shoot in Badami; the expression of osa-miR12480 increased in shoot and decreased in root, whereas that of osa-miR12482 decreased in shoot and increased in root.

## Northern blot analysis of novel miRNAs in halophytes

The possible involvement of the novel miRNAs in salt tolerance was tested by looking at their expression in two halophytes, *O. coarctata*, a wild rice, and *S. maritima*, a plant growing along sea shore (Fig 3, S6 File). Expression of seven out of the nine novel miRNAs, including miR12475, miR12478, miR12481, miR12482, miR12477, miR12479 and miR12480, could be detected in the wild rice *O. coarctata* (Fig 3). The expression of miR12475 and miR12477 was upregulated in response to salt application in *O. coarctata*, although the upregulation of miR12475 was only a little compared to that of miR12477. The expression of miR12478, miR12479, miR12480, and miR12481 on the other hand was downregulated greatly in the plant upon salt treatment. The expression of miR12482 also showed a little down regulation in response to the salt treatment. In contrast to *O. coarctata*, however, only two of the novel miR-NAs, miR12477 and miR12482 expressed in the halophyte *S. maritima* grown in absence of NaCl (control), and their expression increased slightly in response to the salt application, simi-lar to that in *O. coarctata* (Fig 3).

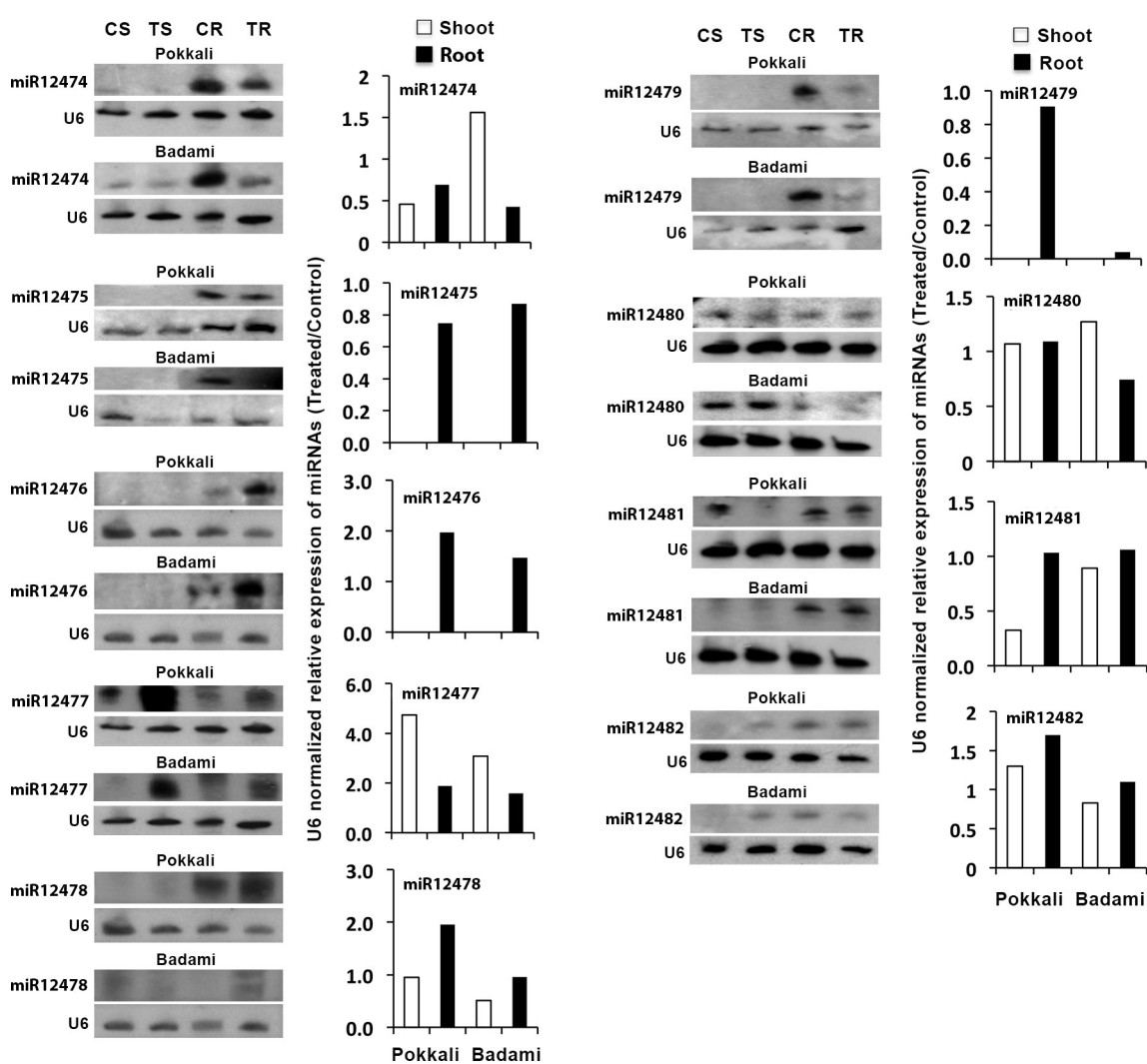

**Fig 2. Northern blots showing expression of the novel miRNAs.** Other details as in Fig 1.

## Validation of target cleavage by 5'RACE, RT-qPCR and dual-luciferase assay

The targets of all the novel miRNAs validated for expression by Northern were predicted using the online software (S7 File). However, since only the novel miRNAs osa-miR12482 and osa-miR12477 expressed in the halophyte *S. maritima*, their target genes were experimentally determined by 5'RACE. Cleavage of three targets, including monocopper oxidase-like protein (*MCO*), L-ascorbate oxidase (*LAO*), and *Myb* could be confirmed by RLM-RACE. *LAO* and *MCO* were targeted by miR12477 and *Myb* by osa-miR12482 (Fig 4). The cleavage of the targets by the miRNAs was validated further by dual-luciferase assay (Fig 5). Significant decrease in F-luciferase to R-luciferase activities ratio was obtained for the target miRNA sequences (Fig 5A) suggesting that *LAO* and *MCO* were targeted by osa-miR12477 and *Myb* was targeted by osa-miR12482. The result was also supported from decrease in expression of F-luciferase in response to the salt treatment (Fig 5B). Expression studies of *LAO* by RT-qPCR showed decrease in its expression in both Pokkali and Badami upon NaCl treatment, comparatively more in the latter than in the former (Fig 6). The expression of Myb, however, showed an insignificant change in Pokkali, but significant increase in root in Badami in response to the salt treatment.

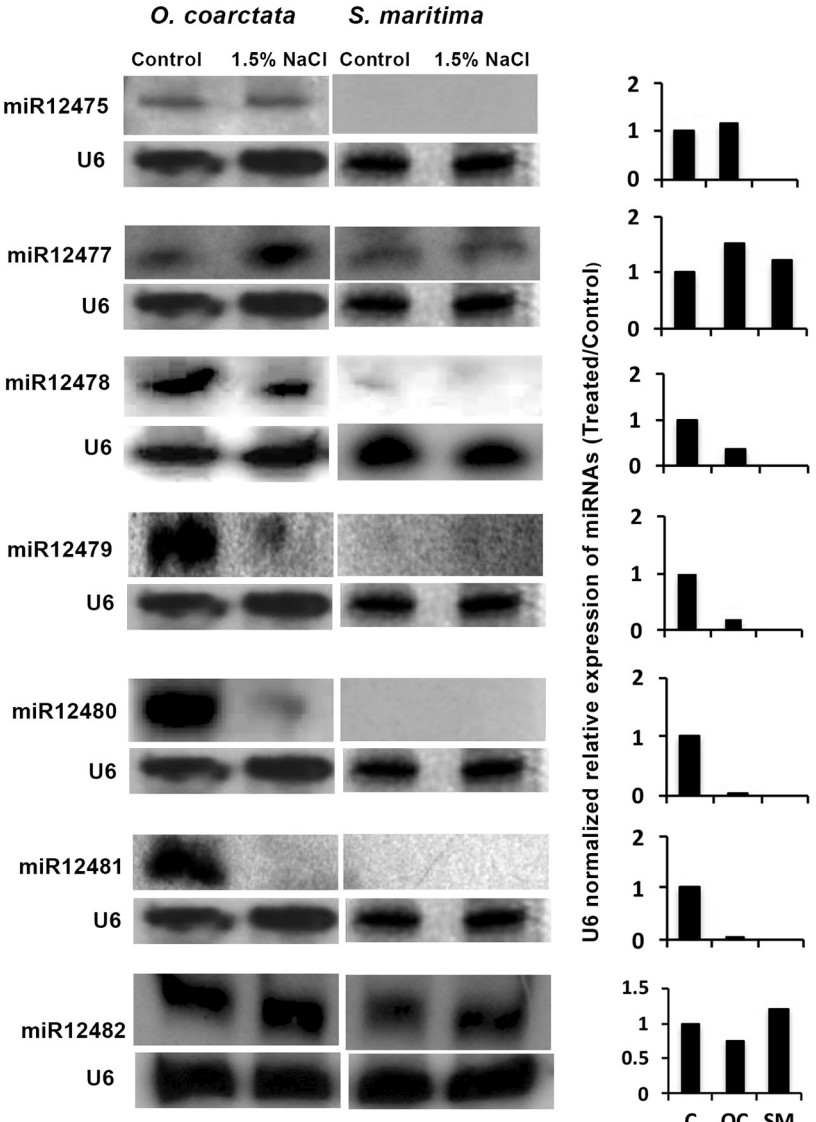

**Fig 3. Northern blot showing expression of the novel miRNAs in the shoot tissue of control (C) and 256 mM NaCl treated (T) *Oryza coarctata* (OC) and *Suaeda maritima* (SM).** NaCl treatment was for 9 h. U6 served as internal control. The right panel shows the relative expression of the miRNAs in the treated over control after normalization of the band intensity with U6.

## ROS accumulation studies

DAB treatment revealed high level of accumulation of reactive oxygen species (ROS) in the leaves of both Pokkali and Badami in response to 256 mM NaCl treatment (Fig 7). However, the accumulation was much greater in the salt-sensitive Badami compared with that in the salt-tolerant Pokkali.

## In situ expression studies of osa-miR12477

In-situ PCR was performed in order to understand the spatial distribution of osa-miR12477 targeting *LAO*. The sections showed expression of the miRNA mostly in the mesophyll tissues, and the leaf section from the NaCl treated seedling showed a greater staining compared to that

**>LOC_Os05g38410.1|12005.m08020|cDNA L-ascorbate oxidase (*LAO*), osa-miR12477 target**

```
CCAAGUAGUUGCGACGUGAGUU
::  :::::::::::::::::::
GGCUCAUCAACGCUGCACUCAACGACGAGCTCTTCTTCTCCATCGCCAACC
              ↑5/6
```

**>LOC_Os11g48060.1|12011.m08604|cDNA monocopper oxidase (*MCO*), osa-miR12477 target**

```
CCAAGUAGUUGCGACGUGAGUU
::  ::::::::::::::::  ::
GGAUCAUCAACGCUGCACUGAATGATGACCTATTTTTCAAGGTTGCTGGGC
              ↑3/6
```

**>LOC_Os07g43420.1|12007.m08570|cDNA myb (*Myb*), osa-miR12482 target**

```
UUUUCUUUAUGGGUUAUAGAACGA
:  ::  ..::::::::: :::::::.:
ACAAUGGAUACCCAAAAUCUUGUUGCTAAAACTGCAGCAAAACAATTATTTGC
              ↑3/6
```

**Fig 4. The cleavage points of L-ascorbate oxidase (*LAO*) and monocopper oxidase (*MCO*) targeted by the novel miRNA osa-miR12477, and of *Myb* targeted by the novel miRNAs osa-miR12482.** The cleavage points, indicated by arrows, were obtained by cloning and sequencing of the 5'RACE PCR products. The numbers against each arrow indicate the number of clones that were having cleavage point, i.e. the insert ligated to the 5' adaptor out of the total number of the clones of the PCR products sequenced.

from the control one (Fig 8). The negative controls (minus the stem-loop RT primer) did not show any staining.

## Discussion

Regulation of gene expression by miRNAs has emerged as an important molecular process determining functioning of several metabolic pathways, including the biochemical events involved in salt tolerance in plants [20,21,40]. This is also reflected from the changes in the frequency of non-redundant reads (S1 File) and the number of unique reads representing miRNAs (Table 1) in both root and shoot in response to challenge of the Pokkali rice seedlings with 256 mM NaCl. Besides, the response of root and shoot to the NaCl exposure also differed, as is reflected from the changes in the predictable miRNAs reads per million; it increased in root but decreased in shoot, suggesting that the changes in the metabolism operational in root and shoot in response to the salt treatment might be different. The same is also reflected form the response of the total miRNAs expression in the two tissues (Table 1, S2 File). Furthermore, a much greater percentage of the unique sRNAs in 24-nt length category than in 21-nt length category (S1 File) suggested the presence of DCL3 action that matures 24-nt sRNAs in angiosperms, unlike in conifers that lacks DCL3 [41], and that these might be derived from transposons and retrotransposons constituting as much as 75% of the total genome with diverse sequences [42], unlike the 21-nt sRNA representing canonical miRNAs [43,44]. In addition, a low percentage of 20–22 nt unique sRNA sequences together with high sequence redundancy in comparison to that of 24-nt unique sequences (S1 File) is indicative of derivation of 20–22 nt sequences from a much lower number of genetic loci representing highly expressing genes, and hence these could be actively involved in regulation of gene expression, most likely as miRNAs [43]. With regard to influence of salinity on sRNA population, the increase in frequency of the unique sequences in root in response to NaCl suggested that the processes leading to accumulation of sRNA are hastened in presence of NaCl, although all of these might not be miRNAs [44].

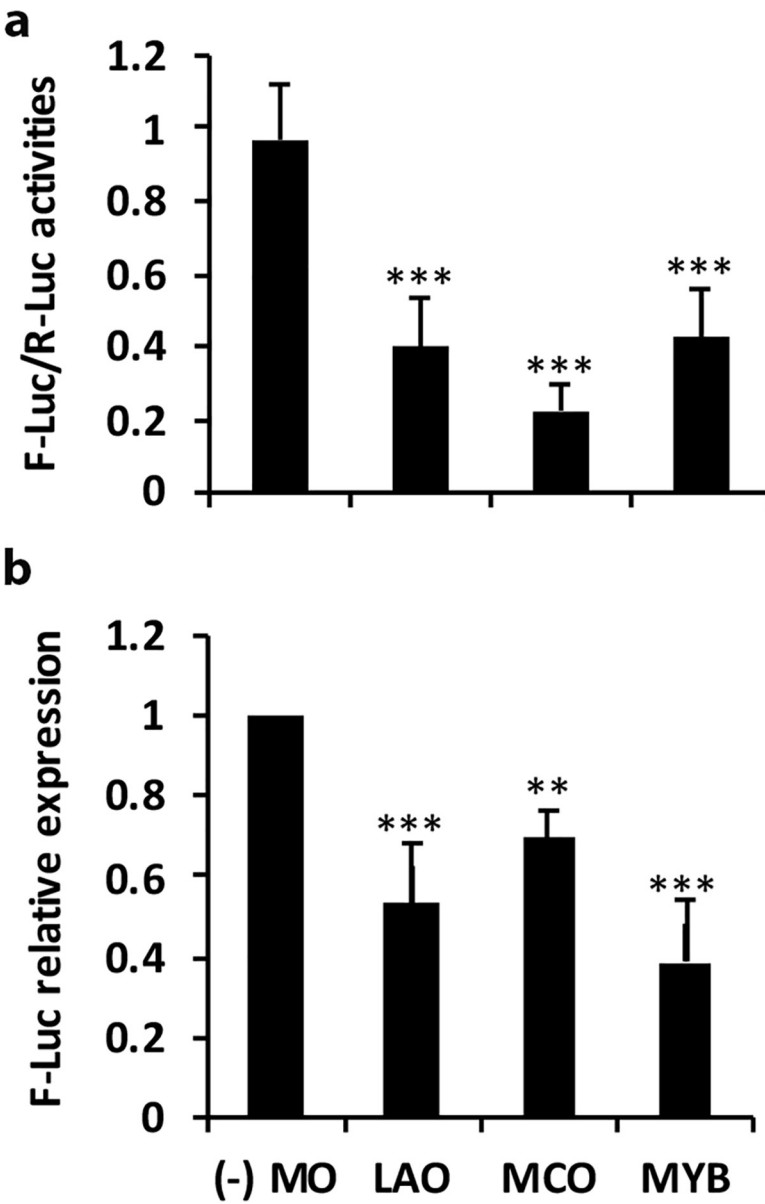

**Fig 5. Dual-luciferase assay for the validation of the target cleavage.** The target sequence of the miRNAs in the three gene products, i.e. L-Ascorbate oxidase (*LAO*), monocopper oxidase (*MCO*), and *Myb* were introduced individually in the 3'UTR of luciferase (Addgene ID 55206). These constructs served as 3'UTR sensor. Construct was also made for overexpression of osa-miR12477 and osa-miR12482 by introducing their hairpin region flanked with 100 bp upstream and downstream sequences separately into Addgene ID 55208 vector. These were termed miRNA overexpressor. The 3'UTR sensor and miRNA overexpressor were then individually introduced into *A. tumefaciens* strain LBA4404. Nine leaves in taking three plants were infiltrated with equal volumes of miRNA overexpressor and 3'UTR sensor. Leaves were also infiltrated only with 3'UTR sensor, referred as (-) MO. These leaves were collected after 48 h and Renilla (R) and Firefly (F) luciferase assays were performed. The ratio of F-Luc/R-Luc activities for the individual 3'UTR sensor and miRNA overexpressor pairs were calculated and presented (a). Parallelly RNA was also extracted from the infiltrated leaves. RT-qPCR was performed for R-luciferase and F-luciferase using primers against these transcripts to see the relative expression of F-luciferase using the expression of R-luciferase as reference (b). The data are mean ± sd of assays in nine individual samples. The asterisk, ** or *** marked against the individual genes indicates that their activities/expressions in the leaves infiltrated with both 3'UTR sensor and miRNA overexpressor for the individual gene and miRNA pairs were significantly less than in the leaves infiltrated only with 3'UTR sensor at $p \leq 0.01$ and $p \leq 0.001$, respectively.

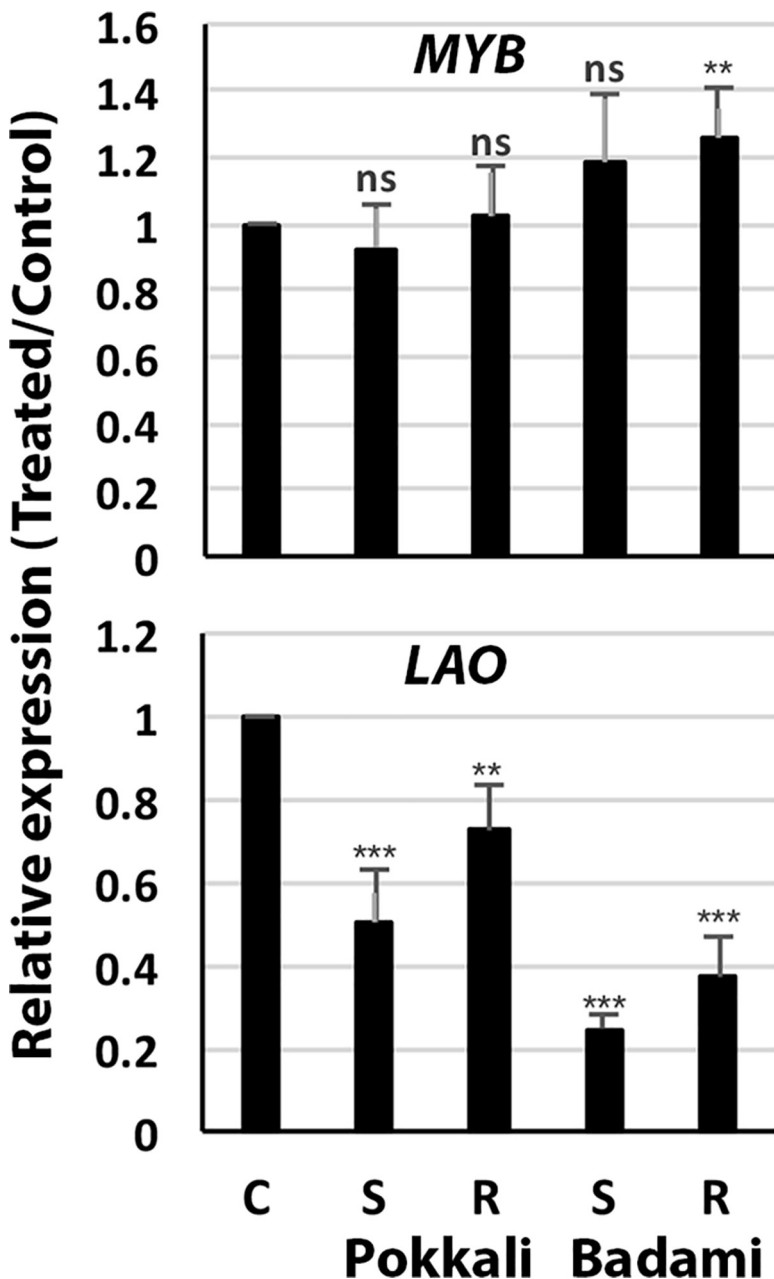

**Fig 6. Changes in expression of *LAO* and *Myb* in shoot (S) and root (R) tissues of *O. sativa* cv. Pokkali and *O. sativa* cv. Badami upon exposure to 256 mM NaCl for 9 h on the 9ᵗʰ day of germination.** The expression was measured by RT-qPCR and expressed as relative value in the treated samples over the control ones considering actin as the reference gene. LAO- L-Ascorbate oxidase, Myb- Myb transcription factor.

The family diversity of miRNAs observed in the current study in terms of the number of miRNAs vis-à-vis the number of miR-families to which they belong, which is approximately 1.7 for shoot and 3.2 for root (S2 File, S3 File), is similar to that obtained in species like *Citrus trifoliate*, *Brassica napus*, cucumber, potato and water melon showing ratio ranging from 1.5 to 2.0 [45–49]. Such low diversity is expected for a plant species with known genome unlike that in plant species with unknown genome, like *Avecinia marina* and *S. maritima* showing miRNAs to miR-families ratio of 5.0 or more [21,38]. However, the number of conserved

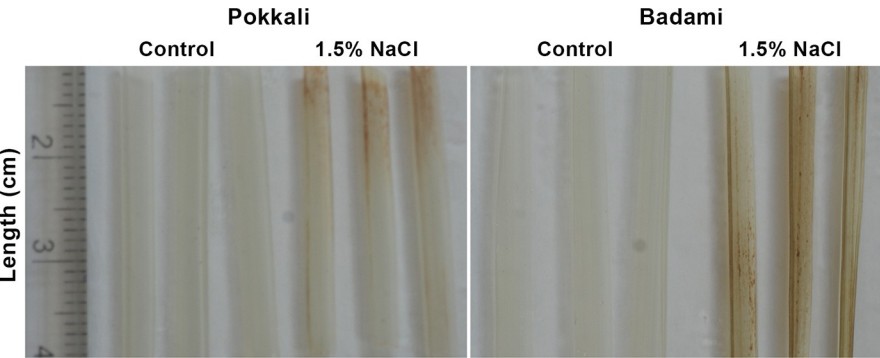

**Fig 7. Influence of NaCl treatment on accumulation of H₂O₂ (ROS) in the leaves of salt-tolerant *O. sativa* cv. Pokkali and salt-sensitive *O. sativa* cv. Badami.** The leaves of the control seedlings and those treated with 256 mM NaCl for 9 h on the 9th day of germination were harvested and processed for DAB staining for the detection of ROS accumulation. Three leaves each from the control and NaCl treated seedlings of both Pokkali and Badami were considered for DAB staining.

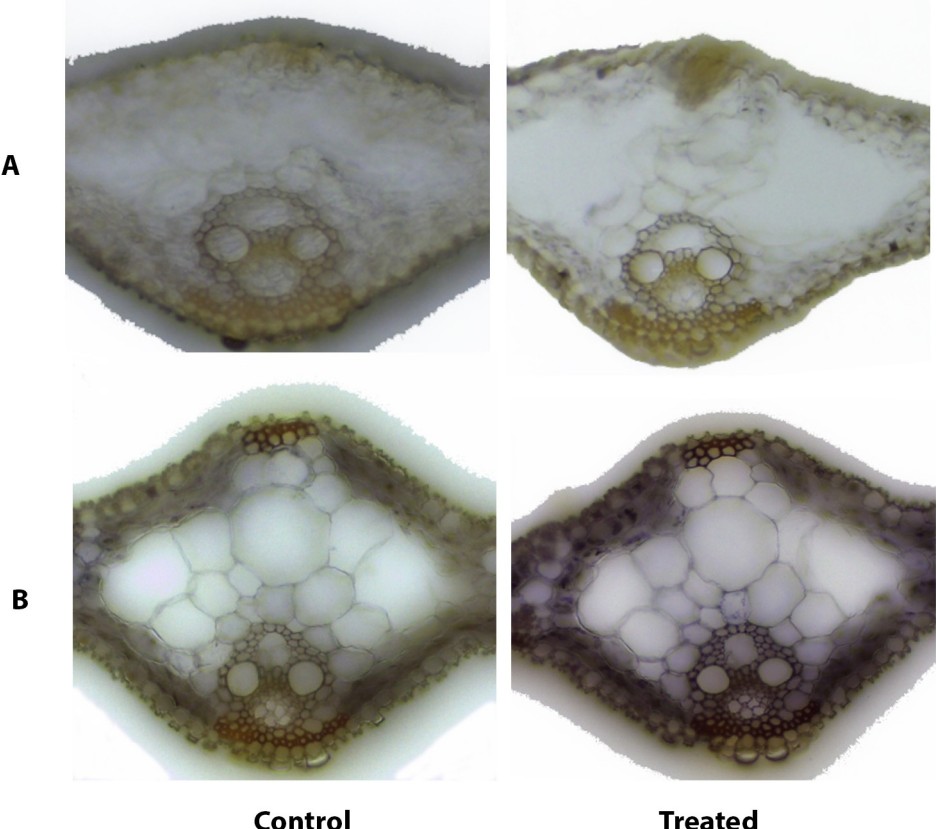

**Fig 8. In situ PCR for the localization and expression studies of osa-miR12477.** Paraffin embedded leaves of the control (C) and 256 mM NaCl treated (T) Pokkali seedlings were sectioned (60 μm) on a microtome, processed to remove the paraffin, osa-miR12477 was reverse transcribed using miRNA specific primer and amplified with miRNA specific stem-loop forward primer and universal reverse primer using high fidelity Taq polymerase with dig-11-dUTP (Roche). The section was probed with anti-DIG-AP-Fab fragment and the miRNA expression was visualized using BM-purple substrate (Roche) on a light microscope. The leaf sections of both control (C) and NaCl treated (T) seedlings were processed without (A) and with (B) osa-miR12477 specific stem-loop primer.

miRNAs found was only 49, and those already reported from rice were only 39 (S8 File), which is in contrast to the reports of presence of as many as 738 miRNAs from rice in miRBase 22.1, suggesting that their expression varies greatly depending upon the cultivar/varieties and growth stage. Moreover, identification of precursors of as many as 10 conserved miRNAs in the rice genome, not reported before, suggested that the miRNAs database for rice is still not saturated. This is also reflected from the fact that the precursors of at least two novel miRNAs, sma-miR5 and sma-miR6, identified in a halophyte *S. maritima* were found to be present in rice and matured miRNAs were salt-responsive [21].

The presence of salt seems to favour the growth of Pokkali. This is reflected from the salt-induced downregulation observed in the NGS result (S2 File) in shoot of the members of miR396, like miR396a/b, which targets Growth Regulator Factors (GRFs) promoting cell proliferation during leaf development [50]. However, the salt-induced appearance of one of the member of the family, miR396g/h, is in opposed to its salt-mediated suppression in *Salicornia europea*, but of course on long-term (7 d) treatment [37]. Hence, the salt-induced appearance of mir396g/h in short-term exposure in the present case (S3 File) might be to downregulate ubiquitin-specific proteases (UBPs), the target of miR396g/h [51], to promote ubiquitin mediated degradation of the undesired proteins during the initial period of salt stress. In contrast to others, the members of miR167 has been reported to be drought responsive [52], and Kinoshita et al. [53] have reported their downregulation under high osmotic stress, in agreement with the present observation in shoot (S2 File) considering the fact that salt application also creates physiological dryness like situation, at least during the initial phase of treatment. Moreover, the targets of mir167, like ARF and IAA-Ala, have also been reported to accumulate under drought or osmotic stress [53,54]. It is well established that ARF class of transcription factors regulates diverse metabolic and developmental process under abiotic stress for the benefit of the plant [39,55], and downregulation of miR167f in response to NaCl application in the root and shoot tissues of Pokkali, but not in Badami (Fig 1), might be enabling Pokkali to tolerate salinity better than Badami. Downregulation of miR159a in root and shoot of Pokkali in response to NaCl, similar to that of miR167f, is also reflective of greater tolerance of the cultivar compared to Badami in which miR159a was upregulated, at least in shoot (Fig 1). This is because miR159 targets Myb transcription factors [39,55,56] that participate in diverse biological processes, including defense and stress responses in plants [39,55,57,58]. In fact, overexpression of *Myb* transcription factors of several plants, like *AmMYB1* and *TaMyb73*, have been shown to improve salt tolerance in tobacco and *Arabidopsis*, respectively [59,60]. In addition, the salt-induced upregulation of the miRNAs targeting transcription factors could be equally important in providing resistance to salinity to the plant [39]. This is reflected from the upregulation of miRNA164a/b/f, miR319a, miR169c and 172a (S2 File, S3 File) targeting *NAC*, *TCP*, *NF-YA* and *AP2/EREBP* domain protein, respectively. It has been seen that overexpression of *NAC* leads to downregulation of several drought-responsive genes, suggesting *NAC* to function as negative regulators of drought tolerance [61]. Similarly, *NF-YA* has been seen to function as negative regulators of salt tolerance, as overexpression of *GmNFYA3* in soybean increases its sensitivity to salinity [62]. Low level of TCP on the other hand increases cell proliferation [63,64], and hence decrease in its level because of salt-induced upregulation of miR319a might be helpful for the plant to maintain cell division and cell growth under the short period of salt stress condition. Similar to TCP, the target of miR172a, AP2/EREBP domain protein SSAC1, is beneficial to plant at low level with regard to salt tolerance, as it functions as transcriptional repressor [65]. The overexpression of miR172a in fact improves salt tolerance in soybean [65]. The members of the family miR172 have also been found to be salt responsive in the wild rice *O. coarctata* and the African rice *O. glaberrima* [39,55], similar to this study.

Although the current database of plant miRNAs for rice is quite rich, it is going to increase further with the discovery of the miRNAs expressed only under certain environmental conditions, stage of development or in specific cultivars/varieties. This is evident from the present finding of at least nine novel miRNAs (Table 2), including five with miRNA* sequence. Although detection of miRNA* adds weight to the authenticity of the predicted miRNA candidates, mostly these may not be detected in sequencing as these are degraded soon after being exported to cytosol, and this makes their abundance in cell much lower than that of their corresponding mature miRNAs [25]. The negative MFE of the precursor miRNAs varied significantly with a range of -34.1 kcal/mol to -58.9 kcal/mol, which are much lower than the values of the tRNA and ribosomal RNA [66]. The mean MFEI value of the precursors was higher than 0.64, the MFEI of tRNAs, the maximum among the known RNAs, reflecting that these newly identified miRNAs were likely to be miRNAs precursors, and not any kind of non-coding RNAs [67]. Besides, the length of the precursors was greater than the desired length of 60-nt, the miRNAs occupied only one of the arm of the concerned secondary structure, and the mismatches between miRNA/miRNA* duplex was not more than five (Table 2, S5 File), suggesting that the precursors could in fact be diced to mature miRNAs [66]. The A+U contents of the precursors varied from 37% to 71.4% (Table 2), in agreement with the reports available [68,69]. The low abundance of a few of the miRNAs may be questionable as per the criteria of qualification by their number, which is generally 10 or more [70]. However, the abundance of non-conserved miRNAs could be at a low level as these might have only specific function at certain stage of development or environmental condition [71].

Salt-responsiveness of the novel miRNAs considered for validation varied greatly not only within the variety/cultivar but also between the varieties/cultivars, as revealed by Northern blot analysis (Fig 2) suggesting their varying role is salt tolerance. The downregulation of the novel miRNAs osa-miR12474, osa-miR12475 and osa-miR12479 in roots of both the rice varieties suggested their positive regulatory role in salt tolerance. In contrast to miR12474 and miR12475, the upregulation of miR12476 and miR12477 in roots in response to the salt treatment (Fig 2) suggested that these novel miRNAs might be having negative regulatory role in salt tolerance in the plants working by downregulating the expression of their target gene(s). The result of Northern for miR12478, miR12480, miR12481 and miR12482 (Fig 2) on the other hand revealed that regulation could be negative or positive depending upon the tissue and the varieties/cultivars. However, for all these miRNAs, Pokkali showed an increased expression in root (Fig 2), suggesting that for salt tolerance downregulation of their target was necessary in root. The downregulation of miR12481 and miR12478 in shoot on the other hand suggested that upregulation of their target in shoot might be necessary for salt tolerance in both Pokkali and Badami, although they differ in salt-sensitivity. Moreover, the Northern data also revealed that upregulation of miR12482 and miR12480 in both root and shoot could be very important for salt tolerance, as their level got upregulated in Pokkali in response to NaCl, but their response to NaCl was not uniform in root and shoot of Badami (Fig 2).

Salt responsiveness of the novel miRNAs like miR12477, miR12478, miR12479, miR12480, miR12481 and miR12482 in the wild rice salt-tolerant *O. coarctata* (Fig 3), particularly the drastic downregulation of miR12478, miR12479, miR12480 and miR12481 is suggestive of their possible role in salt tolerance. Expression of only miR12477 and miR12482 in *S. maritima* suggested that the other miRNAs are conserved mainly in the related species, and that the molecular mechanism regulating the salt tolerance process may differ among species taxonomically placed very apart.

Cloning and sequencing of the RACE PCR products (Fig 4) and luciferase assay (Fig 5) confirmed *LAO* and *MCO* as the targets of osa-miR12477, and *Myb* of osa-miR12482. The importance of Myb transcription factors in diverse biological processes, including defense and stress

responses in plants is well known [39,55,57,58]. In fact, overexpression of *Myb* transcription factors of several plants, like *AmMYB1* and *TaMyb73*, have been shown to improve salt tolerance in tobacco and *Arabidopsis*, respectively [59,60]. In contrast, the role of *LAO*, targeted by miR12477, is not well established, although its substrate, ascorbate, is a well-known antioxidant, as it reacts with $H_2O_2$ to break it down into water [72]. The reaction, nevertheless, is facilitated by ascorbate peroxidase [73,74]. Furthermore, it is also well established that oxidative stress is one of the way by which salt produces its toxic effect in plant [74]. Hence, downregulation of *LAO* as a result of upregulation of miR12477 in response to salt application suggests it to be a negative regulator of salt tolerance in plant. The negative relationship between *LAO* expression and osa-miR12477 expression is clearly evident from an increased accumulation of osa-miR12477 (Fig 8) concomitant with decrease in expression of *LAO* (Fig 6) in leaf of Pokkali in response to exposure of the seedlings to NaCl. However, the downregulation of *LAO* may not be the sole factor of salt tolerance in plant as its downregulation in response to salt treatment was significantly greater in the salt-sensitive Badami than in the salt-tolerant Pokkali (Fig 6), but ROS accumulation was more in the former than in the latter at 1.5 NaCl treatment, as revealed by DAB staining (Fig 7). A greater accumulation of ROS in Badami than in Pokkali despite a greater downregulation of *LAO* in the former than in the latter could be because of poor functioning of the enzymes involved in scavenging ROS, like catalase, ascorbate peroxidase and peroxidases [74]. It is well known that salt tolerance is a quantitative trait involving many genes. The result of in situ expression of osa-miR12477 showing an increase in expression of the miRNA in Pokkali in response to NaCl treatment of the seedlings (Fig 8), nevertheless, does support an important role of osa-miR12477 in oxidative stress management in the plant.

## Conclusions

The study thus led to identification of several salt-responsive novel and conserved miRNAs, and expression of several of them could be successfully validated by Northern blot analysis. Among the conserved miRNAs, precursors of several of them were different from those available at miRBase 22.1 for rice, and hence these precursors could be considered as novel (S5 File). Several of the conserved miRNAs expressed only in shoot in response to salt, like miR172a and miR393a, targeting genes encoding proteins like AP2/EREBP and auxin receptor in SCF-E3 ligase indicating the involvement of ethylene and auxin in salt tolerance, and more importantly the modulation of hormonal action and signaling in the salt tolerance process. The fact that salt tolerance in plants could be auxin mediated also stems from 1) upregulation of miR171b (S2 File, S3 File), which targets the gene scarecrow-like proteins that negatively regulates GA/auxin action, 2) downregulation of miR167f in shoot in Pokkali (Fig 1) and sma-miR7 in the halophyte *S. maritima* [21], both targeting *ARF*, and 3) the reports of accumulation ARF and IAA-Ala under abiotic stress [53,54]. The second line of miRNA mediated salt tolerance in plants could be through influence on transcription factors regulating expression of genes encoding proteins involved in a wide range of biochemical and physiological processes linked to abiotic stress tolerance without the involvement of plant hormones, like NAC, NFYA, TCP, HD-Zip III and Myb, targeted by miR164a/b/f, miR169c, miR319a, miR166 and miR159a, respectively. The study further revealed that the tolerance to salinity could also be a result of mitigation of oxidative damage that abiotic stress generally inflicts in plants [74], as was reflected from 1) a high salt-induced upregulation of the novel miRNA miR12477, comparatively more in Pokkali than in Badami (Fig 2), that targets *LAO* (Fig 5), the enzyme that breaks down the ascorbate, an $H_2O_2$ detoxifying agent, and 2) low accumulation of ROS in the leaves of Pokkali than in the leaves of Badami upon exposure of the seedlings to NaCl, as

reflected by DAB staining of the leaves (Fig 7). Thus, salt tolerance process in plant is not only complex, being determined by multiple factors, the involvement of miRNAs increases the complexities further. Nevertheless, a clear understanding of regulatory role of miRNAs in salt tolerance may provide opportunities for biotechnological interventions for improving salt tolerance in the crop cultivar of interest, as it would be possible to target the top biochemical and molecular events determining the salt tolerance.

## Materials and methods

### Plant growth conditions and treatment

Seeds of the salt tolerant *Oryza sativa* L. ssp. *indica* cv. Pokkali [75,76] and salt-sensitive *O. sativa* ssp. *indica* cv. Badami [77] were collected from the National Rice Research Institute (NRRI), Cuttack and Orissa University of Agriculture and Technology (OUAT), Bhubaneswar, respectively who maintain the germplasm of the stated rice cultivars. The seeds were surface sterilized, soaked overnight and spread onto a large petri-dish containing moist tissue paper to aid germination. The germinated seeds were grown for 8 days in Hoagland's medium in a plant growth chamber maintained at 25 ± 2 ˚C and ~70% relative humidity with 14/10 h light/dark cycle. Salt treatment was given on the 9th day in morning as 256 mM (1.5%) NaCl [21,77]. EC of 256 mM NaCl is approximately 21 mS/cm [78]. At first a known volume of NaCl stock solution was added to the individual beakers to raise the salt concentration to approximately 85 mM, a low initial NaCl concentration to avoid sudden stress shock. Light was switched on after 30 min of salt application followed by addition of more of NaCl stock solution (volume calculated beforehand considering the volume of the space up to the net) and 1/10th Hoagland solution up to the net to bring the final NaCl concentration to 256 mM. After exposure of the seedlings for 9 h under illumination the shoot and root tissues of both control and salt-treated seedlings were harvested and immediately frozen in liquid nitrogen and stored at -80 ˚C for future use.

*Suaeda maritima* (L.) Dumort. seeds were collected from the plants growing naturally along the coastal belt of Bhadrak, Odisha. The species was identified by Dr. Pratap Chandra Panda, Taxonomy and Conservation Biology Division at the Regional Plant Resource Centre (RPRC), Bhubaneswar. The species is grown and maintained at the Institute of Life Sciences, Bhubaneswar at its Green House facility [21,79], and has also been deposited in the publicly available herbarium of RPRC (Voucher no. 10693). The seeds of the plant were germinated over autoclaved soil in plastic pots having holes at the bottom. The pots were placed in a growth chamber maintained at 25 ± 2 ˚C and ~70% relative humidity, and were illuminated with white fluorescent light (~200 μmol m$^{-2}$ s$^{-1}$) at 14/10 h light/dark diurnal cycle. Hoagland's solution (1/10th) or Milli-Q water were used alternately to water the pots. The seeds germinated and the seedling grew up to 2–3 cm in 3–4 weeks. The seedlings were then transferred to other similar pots with 2–3 seedling in each and were grown under natural day/night cycle for ~3 months in a greenhouse maintained at 25 ± 2 ˚C and ~70% relative humidity. Milli-Q water and 1/10th Hoagland's solution were used to water the pots alternately every day. On the day of NaCl application the individual pots meant for treatment received 500 mL 85 mM NaCl prepared in 1/10th strength and the control pots received only 1/10th Hoagland's solution. The excess solution got drained away through the pores. The initial treatment with 85 mM NaCl was given to avoid salt stress shock to the plants. After 30 min the treatment pots received 100 mL of 256 mM NaCl prepared in 1/10th strength Hoagland's solution and the control pots received only 1/10th Hoagland's solution. The process was repeated every hour. The above ground part of the seedlings, the shoot, from the control and treated pots were harvested after 9 h and were preserved in liquid N$_2$ for use later on.

The wild rice *Oryza coarctata* (Roxb.) Tateoka was collected from the lake Chilika outer channel, Puri, Odisha. The species is grown and maintained at the Institute of Life Sciences, Bhubaneswar at its plant growth facility [21]. The taxonomic details of the species was provided by Dr. Pratap Chandra Panda, Taxonomy and Conservation Biology Division at RPRC, and the species has also been deposited in its publicly available herbarium (Voucher no. 10689). Young plants were raised by rhizome transplantation in green house in pots similar to that used for *S. maritima*. The young plants, 15–20 days old, grown in different pots were then given NaCl treatment in a way similar to that to *S. maritima*, and the aerial part of the control and treated plants were harvested and stored frozen for use in analysis.

## Small RNA library construction, Illumina sequencing and bioinformatics

For the identification of the salt-responsive miRNAs, sRNA sequencing of control (C) and treated (T) samples of root (R) and shoot (S) of Pokkali was performed on Illumina platform. Total RNA was extracted from the individual samples using TRIzol reagent (Thermo Fischer Scientific) following manufacturer's instructions. Only good quality RNA with RIN value above 7.0 was used for downstream processing. Total RNA (1 μg) from each sample was subjected to electrophoresis on denaturing polyacrylamide gel to separate out the sRNA fragments, which were recovered and ligated with 3'- and 5'- adaptors. The ligated sRNA products from each sample were reverse transcribed and PCR amplified to generate the sRNA libraries. The individual libraries were normalized to 2 nM with Tris-HCl as per TruSeq Small RNA Prep kit manual (Illumina). The libraries were then processed for single-end sequencing on HiSeq2000. The sequencing results were generated as FASTQ data using Illumina pipeline CASAVA 1.8 package. NGS QC Toolkit Cutadapt and UEA Small-RNA Workbench were used to process the raw reads. Adaptor-trimmed high quality reads were queried against the NCBI and Rfam databases to screen out and discard the abundant non-coding RNAs like tRNA, snoRNA, snRNA and others. The unique reads were searched for the known sequences in the miRBase 22.1 for the identification of the conserved miRNAs. The sRNA sequences were then mapped on rice genome for the identification of possible novel miRNAs and their precursors. The basic criteria described in literature were followed for putative miRNA prediction [67,80–82].

## Expression studies of miRNAs by Northern blotting

Expressions of the conserved miRNAs identified and that of the novel miRNAs predicted bioinformatically were validated by Northern blot analysis. Both shoot and root tissues of the control and 256 mM NaCl treated plants were considered for the validation of their expression. This also allowed study of their tissue-specific and NaCl-induced differential expression. Total RNA was extracted from different tissues using TRIzol. For the Northern blot analysis 10 μg of good quality RNA (260/280 ~2.0) of the individual samples was loaded in different lanes of 15% PAGE and run at 200 V. The RNA separated was transferred on a nylon membrane using Trans-Blot® SD Semi-Dry Electrophoretic Transfer Cell (Bio-Rad), UV-cross linked at 150 mJ using a UV cross-linker (Hoefer™ UVC 500 Crosslinker) and was stored at 4 ˚C until used for hybridization. The individual membranes were pre-hybridized in hybridization buffer (Sigma) at 37 ˚C for 1 hour separately in hybridization bottles. DNA oligos complementary to the individual miRNAs were labeled with [γ-$^{32}$P]dATP using T4 polynucleotide kinase (Thermo Fischer) and used as probes. The individual labeled probe was allowed to hybridize separately with the pre-hybridized blots by incubation for 16 h at 37 ˚C in hybridization bottles. The membranes from the individual hybridization bottles were washed separately with 2X SSC buffer containing 0.1% SDS (non-stringent), followed by washing with 1X SSC buffer

containing 0.1% SDS (non-stringent) for 15 min each at 37 ˚C. The membranes were dried and exposed to X-ray film individually for appropriate time. The X-ray films were developed and fixed, and the images were analyzed on a gel-doc. Each exposed membrane was then sequentially washed with stringent and non-stringent wash buffers for 10 min each at 80 ˚C to strip off the probe. The stripped membranes were air dried and probed with a small stretch of U6 RNA labeled with [$\gamma$-$^{32}$P]dATP, washed and exposed to X-ray film, as stated above, for detection of the signal. U6 RNA signals served as internal control. The signals of the miRNA and that of U6 signal on the same blot were imaged on gel-doc (BioRad) and analyzed taking densitometric approach. The expression of the individual miRNAs in the control and NaCl-treated samples was shown as relative value after normalization with the respective U6 expression. List of the probes used are given in the S9 File.

## Prediction of targets

The targets for the experimentally validated miRNAs were searched in the rice transcriptome database using psRNAtarget software (http://plantgrn.noble.org/psRNATarget/) with default parameters. A single miRNA was bioinformatically found to have a number of targets, but the targets with expectancy $\leq$ 3.0 were mostly considered for annotation and validation. Complete CDS of the individual targets was retrieved from the NCBI site and searched for similarity in the *Oryza sativa* L. indica database using BLAST programme at the EnsemblPlants resource database (http://plants.ensembl.org/index.html). The search at the EnsemblePlants resource was to ensure that the target sequence was present in the Indica variety as well [83,84].

## Validation of miRNA targets by RT-qPCR

Real-time PCR (RT-qPCR) was one of the methods that was used to validate the target gene of a miRNA. Total RNA was extracted using TRIzol reagent (Invitrogen, Life Technology) from the root and shoot tissues of the control and NaCl-treated seedlings. The RNA from the individual samples was converted to cDNA using QuantiTect Reverse Transcription Kit (Qiagen). The cDNA so obtained for the individual samples served as template for the qPCR carried out using QuantiTect SYBR Green PCR Kit (Qiagen) following the kit manual. Primers specific to the individual genes and mostly 20-mer or more in length with Tm not less than 59 ˚C were designed using Primer Blast software at NCBI (http://www.ncbi.nlm.nih.gov/tools/primer-blast/). LightCycler® 480II (Roche) machine was used to carry out the qPCR. For each cDNA preparation PCR was run in triplicate, and two cDNA preparations from separate biological samples were used for the determination of expression of each gene (n = 6). The expression level of actin gene (LOC_Os03g50885) in a biological sample was used for normalization of expression of a target gene in that sample. The expression of a target gene was presented as relative value in the NaCl-treated sample compared with that in the control sample following Pfaffl [85] considering the expression of actin as reference. Paired 't' test was used to find out significance of difference in expression of a gene in the NaCl-treated sample compared with that in control.

## Validation of miRNA targets by dual-luciferase assay

Quantification of miRNA mediated target repression was performed using dual luciferase transient expression system. It was performed by transiently expressing the miRNA alongside expressing its target gene in *Nicotiana benthamiana* [86]. *N. benthamiana* plant was grown under controlled environmental conditions for the experiment in a greenhouse. Natural day/light illumination was provided maintaining the temperature of the greenhouse at 24 ± 2 ˚C and relative humidity at 65–75%. Matured leaves from two months old plants were used for

the transient expression of miRNA and the target gene required for dual-luciferase assay. Sense and antisense strands were synthesized for the target oligo of the individual miRNAs with AvrII restriction site introduced as extension at the 5' end and AgeI restriction site introduced as extension at the 3' end. The sense and anti-sense strands (5 µL each, 0.5 nM concentration) were taken in a PCR tube along with 1.67 µL 0.3 M NaCl in a final volume of 60 µL. PCR was programmed for initial temperature of 97 ˚C for 5 min followed by ramp down to 20 ˚C at 0.1 ˚C/s. The annealed product was double digested with AverII and AgeI for cloning into pGreen derived plasmid at 3'UTR of luciferase (Addgene ID 55206). In a separate reaction 1 µg of pGreen plasmid (ID 55206) was double digested with AverII and AgeI and the digested product was run on 1.2% agarose gel for visualization and recovery of the correct fragment for use in cloning. The double digested vector (ID 55206) and the double digested target oligo were mixed in 1:3 ratio and incubated at 16 ˚C for 16 h in the presence of T4 DNA ligase for the ligation to occur. The ligated product was referred as 3'-UTR sensor. For making the miRNA overexpressor construct the precursors of the individual miRNAs, the hairpin region including ~100 bp upstream and downstream sequences, were PCR amplified. The forward primer used had XhoI site and reverse primer had EcoRI site at the 5' end. The PCR product was double digested with XhoI and EcoRI and cloned into pGreen II based vector (Addgene ID 55208). The 3'-UTR sensor and miRNA overexpressor constructs so obtained were transformed into DH5α competent cells, which were plated on agar plates following standard method for the development of colonies. Individual colonies were picked up and subjected to colony PCR and sequencing for confirmation of the presence of inserts. The 3'-UTR sensor and miRNA overexpressor plasmids were transformed into *A. tumefaciens* strain LBA4404 individually, and equal volume of the two was infiltrated together in the leaves of *N. benthamiana*. A total of nine leaves to a maximum of three leaves per plant were infiltrated. *A. tumefaciens* carrying the sensor construct was also infiltrated alone in similar sets of leaves. After 48 h of infiltration the leaves were harvested and frozen in liquid $N_2$. Renilla (R) and Firefly (F) luciferase activities were estimated in the preserved leaf tissues infiltrated with both 3'-UTR sensor and miRNA overexpressor and that infiltrated only with 3'-UTR sensor. Dual-luciferase Reporter Assay System (Promega) was used to perform the luciferase assay using Glomax 20–20 luminometre (Promega). RNA was also isolated from these leaves for the study of the luciferase transcript levels. RNA isolated from the leaves of both infiltrated with 3'-UTR sensor and miRNA overexpressor together and that infiltrated with 3'-UTR sensor alone was reverse transcribed using QuantiTect Reverse Transcription Kit (Qiagen). Primers were designed for the amplification of F-luciferase and R-luciferase and were used for the qPCR. The cDNA preparations from the RNA extracted from the leaves of both infiltrated with 3'-UTR sensor and miRNA overexpressor together and that infiltrated with 3'-UTR sensor alone were used for qPCR. R-luciferase served as the reference gene for the quantification of F-luciferase expression.

## Validation of miRNA targets by 5'-RACE

Ambion RLM-RACE (RLM- RNA Ligase-Mediated; RACE- Rapid Amplification of cDNA Ends) kit (Thermo Fischer Scientific) based on 5'-RACE was used for the miRNA target identification. Total RNA was isolated from the root and shoot of the NaCl treated seedlings, and 10 µg of it was incubated individually with 5' adapter for its ligation at the 5' end of the cleaved RNA, which contains a phosphate group because of the DICER action, following the kit manual. The ligated product (2 µL) was incubated with M-MLV reverse transcriptase, random decamers and other reagent supplied along with the RLM-RACE kit for 1 h at 42 ˚C for reverse transcription to cDNA to make the RACE library. The miRNA cleaved gene product was PCR

amplified taking 1 μL of the concerned RACE library and using 5' RACE adapter-specific and gene-specific outer primers. The PCR product (1 μL) was again subjected to PCR for 30 cycles using 5' RACE and gene-specific inner primers to amplify the concerned gene-product, which was electrophoresed to get the amplicon that was excised and eluted and cloned into pGEMT easy vector. The vector along with the cloned product was transfected into DH5∞ competent cells and the transformed bacteria were grown in 1 mL LB medium at 37 ˚C overnight in a shaking incubator at 200 RPM. The bacteria were pelleted out and spread on LB plate that contained ampicillin. The insert in the individual white colonies was PCR amplified by colony-PCR using SP6 T7 primers and visualized on agarose gel. The band from each colony PCR was excised and sequenced, and a record was made for the number of colonies sequenced and the number of colonies that had the insert of the predicted target gene product.

### Tissue level reactive oxygen species detection by DAB staining

The detection of reactive oxygen species (ROS), i.e. hydrogen peroxide ($H_2O_2$) and superoxide radical ($O_2^-$) was done using DAB (3, 3'-diaminobenzidine) standard staining procedure [87]. DAB solution (0.1%, pH 3.0) was prepared in Milli-Q water containing 0.05% Tween 20 and 10 mM sodium phosphate buffer (pH 7.0), followed by addition of HCl. Approximately 5 cm of the leaves cut from the control and 256 mM NaCl treated seedlings were immersed in the DAB solution in separate 15 mL falcon tubes. The tubes were gently vacuumed to allow infiltration of the DAB solution evenly for 5 min. Further, the tubes were placed on a shaker for 4–5 h shaking at 80–100 rpm. Following incubation, the solution in the individual tubes was replaced with bleaching solution (ethanol:acetic acid:glycerol 3:1:1) and placed in boiling water bath (90–95 ˚C) for 15 min till the leaves were fully bleached. The bleaching solution was replaced with a fresh one and kept at RT for 30 min. The leaves were stretched on glass plate and photographed for DAB staining (dark brown colour).

### Localization of miRNAs in leaf tissue by in-situ PCR

Undamaged leaves of Pokkali from the control and NaCl-treated seedlings were harvested and fixed with FAA (formalin-acetic acid-alcohol) fixative by vacuum infiltration. The fixed samples were carefully embedded in paraffin blocks and sectioned (60 μm) using Leica RM2265 Rotary microtome. The sections were carefully collected in microfuge tubes for successive treatments to remove the paraffin [88]. The samples were treated with DNase. The subsequent reverse transcription reaction for the miRNA was done as described by Varkonyi et al. [89]. PCR was then done using Phusion high-fidelity DNA polymerase (NEB) following the manufacturer's instructions. Dioxigenin-11-UTP (Roche diagnostics, 4 μM final concentration) was used in the above standard PCR as an additional reagent in order to facilitate visualization of the miRNA using specific substrate [90]. The color detection was done using BM-purple substrate (Roche diagnostics) and microscopic visualization was done on Zeiss Axio Imager M2 Microscope (Carl Zeiss, Oberkochen, Germany) under bright field illumination conditions.

### Supporting information

**S1 File. Frequency of non-redundant sequences of sRNA.**
(DOCX)

**S2 File. Family distribution of the conserved miRNAs.**
(DOCX)

**S3 File. Family distribution of the conserved miRNAs with sequences.**
(XLSX)

**S4 File. Uncropped Northern blot images of known and novel miRNAs.**
(PPTX)

**S5 File. Precursor sequences of the conserved and novel miRNA and novel miRNAs hairpins.**
(XLSX)

**S6 File. Uncropped Northern blot images of novel miRNAs in *S. maritima* and *O. coarctata*.**
(PPTX)

**S7 File. Targets of the novel miRNAs.**
(XLSX)

**S8 File. Species distribution of the conserved miRNAs.**
(XLSX)

**S9 File. Probes used in the study for the detection of miRNAs.**
(XLSX)

## Acknowledgments

The work was executed utilizing the laboratory facilities of the Institute of Life Sciences (ILS) for which the authors thanks the Director, ILS. The authors also extend thanks to Dr. Santibhusan Senapati of ILS for his help in microtomy. Finally, the authors are thankful to Dr. A. K. Sinha, National Institute for Plant Genome Research, New Delhi for his help in language editing.

## Author Contributions

**Conceptualization:** Birendra Prasad Shaw.

**Data curation:** Tilak Chandra.

**Formal analysis:** Tilak Chandra, Birendra Prasad Shaw.

**Funding acquisition:** Birendra Prasad Shaw.

**Investigation:** Shaifaly Parmar, Sachin Ashruba Gharat, Tilak Chandra.

**Methodology:** Shaifaly Parmar, Sachin Ashruba Gharat, Tilak Chandra.

**Supervision:** Birendra Prasad Shaw.

**Validation:** Tilak Chandra.

**Writing – original draft:** Birendra Prasad Shaw.

**Writing – review & editing:** Ravichandra Tagirasa, Lambodar Behera, Sushant Kumar Dash, Birendra Prasad Shaw.

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
