## [Decision Letter · Decision Letter 0]

14 Oct 2019

PONE-D-19-25289

Identification and expression analysis of miRNAs and elucidation of their role in salt tolerance in rice varieties susceptible and tolerant to salinity

PLOS ONE

Dear Dr. Shaw,

Thank you for submitting your manuscript to PLOS ONE. After careful consideration, we feel that it has merit but does not fully meet PLOS ONE’s publication criteria as it currently stands. Therefore, we invite you to submit a revised version of the manuscript that addresses the points raised during the review process.

We would appreciate receiving your revised manuscript by Nov 28 2019 11:59PM. To enhance the reproducibility of your results, we recommend that if applicable you deposit your laboratory protocols in protocols.io, where a protocol can be assigned its own identifier (DOI) such that it can be cited independently in the future. For instructions see: http://journals.plos.org/plosone/s/submission-guidelines#loc-laboratory-protocols

We look forward to receiving your revised manuscript.

Kind regards,

Alberto Amato

Academic Editor

PLOS ONE

Journal Requirements:

Additional Editor Comments (if provided):

Reviewers' comments:

Reviewer's Responses to Questions

**Comments to the Author**

1. Is the manuscript technically sound, and do the data support the conclusions?

Reviewer #1: No

Reviewer #2: Partly

2. Has the statistical analysis been performed appropriately and rigorously? 

Reviewer #1: Yes

Reviewer #2: No

3. Have the authors made all data underlying the findings in their manuscript fully available?

Reviewer #1: No

Reviewer #2: Yes

4. Is the manuscript presented in an intelligible fashion and written in standard English?

Reviewer #1: Yes

Reviewer #2: Yes

5. Review Comments to the Author

Reviewer #1: 1.In abstract: “However, despite the salinization issue the production of rice is required to be increased from the current figure of ~500 million tons to ~800 million tons by 2050 to meet the demand of the ever growing population”. Being general statement, it should be deleted.

2.In Materials and Methods: Kindly cite the reference mentioning that how the authors have chosen these two cultivar which it shows have contrast salt adaptive. Pl cite the references for Pokkali and badami as sensitive to salinity.

3.Line # 541: Replace the 1.5% Nacl to EC value which is more acceptable form of salinity unit

4. It is wrong to treat the two halophytes at 1.5 % as in this concentration no stress will be induced in two halophyte. Thus comparing them with rice in the same concentration will be conceptually wrong.

5.The author should cite and discuss the most related following two papers; a) Mondal TK, Ganie SA, Debnath AB (2015) Identification of novel and conserved miRNAs from extreme halophyte, Oryza coarctata, a wild relative of rice. PLoS ONE 10(10): e0140675. doi:10.1371/ journal.pone.0140675and b) Mondal TK, Panda AK, Rawal HC, Sharma T R (2018) Discovery of microRNA-target modules of African rice (Oryza glaberrima) under salinity stress. Scientific Reports 8, Article number: 570

6.What is the need of fragmented salinity treatment described in line number 562-568?

7.Line #654. Which actin gene was used as internal control? . Kinldy write the LOC number.

8.Finally since this is the sequence based research and I could not check the sequence as it was not submitted to NCBI and open for review. This is also against the policy of PLOS one

Reviewer #2: This manuscript represent an interesting investigation that focuses on identification of new salt-responsive miRNAs in two rice varieties, susceptible and tolerant to salinity, respectively. After high throughput sequencing large number of both, new and conserved miRNAs, were identified. Tissue specific expression (in roots and shoots) of some miRNA was further validated by Northern blotting and the targets of the novel miRNAs were determined by 5’RACE, dual luciferase assay and RT-qPCR.

The manuscript is well written and easy to read. Large number of data are presented in 8 figures and 7 supporting files and the obtained results support the final conclusions. Unfortunately, almost all data presented are without the number of replication and without bars for standard error or deviation.

Specific comments.

1. Abstract and discussion sections are long. Abstract should focus on the scope of the work and obtained results rather then on “ever growing population” (row 30) or world rice production (row 29).

2. Results.

a. Why authors decided to express the concentration of applied NaCl in % and not in mM as in the major number of publications?

b. Will be much easier to follow the text if data in Fig. 1 and 3 are presented in accelerating order regarding the miRNAs numbers. In the current version is difficult to find the respective miRNA on the figures. For example miR169 is following miR396 (Fig. 1) and miR12477 is after miR12478 (Fig. 3).

c. Some of the photos on Fig.3 are with bad quality (for example miR12482) and is difficult to accept the conclusions for the relative expression level. Definitely, some of the Northern must be repeated or (if available) better photos should be presented. Again, on Fig.3 differences in the expression levels of miR12475 and miR12477 are minimal and not sufficient to claim up-regulation in O. coarctata (row 249, 250). In this case, to enforce their conclusions, authors have to specify the number of Northern experiments that were done for each miRNA.

d. Fig. 6. What mean the control sample with relative expression of 1? The normal way of presentation of RT-qPCR results is 2-ΔCT.

e. Fig 7. On the upper panel scale bar is missing to confirm that the stained leaves have the same size. In this case, instead of the one dimensional and very rough densitometrical analysis, some simple but precise method for quantification of H2O2 content in DAB stained tissue should be applied (for example Ramel et al., 2009). In the lower panel of Fig. 7 the bars for standard error or deviation are missing, together with the total number of stained leaves.

3. Material and methods are long and scholastic for the level of PLOS. For example:

Row 673: The PCR tube was placed in a PCR machine…

Row 724 : The entire PCR product (50 �L) was electrophoresed on 1.5 % agarose gel….

Row 741: dissolving 50 mg in 45 ml… (only the final concentration has to be reported).

Definitely, large part of this section must be revised and shortened.

4. References. Unfortunately, all the references are presented in inappropriate way. According to the “instructions for authors” PLOS uses the reference style outlined by the International Committee of Medical Journal Editors (ICMJE). Example format:

Hou WR, Hou YL, Wu GF, Song Y, Su XL, Sun B, et al. cDNA, genomic sequence cloning and overexpression of ribosomal protein gene L9 (rpL9) of the giant panda (Ailuropoda melanoleuca). Genet Mol Res. 2011;10: 1576-1588.

6. PLOS authors have the option to publish the peer review history of their article (what does this mean?). If published, this will include your full peer review and any attached files.

Reviewer #1: No

Reviewer #2: No

---

## [Author Response · Author response to Decision Letter 0]

18 Feb 2020

3. Have the authors made all data underlying the findings in their manuscript fully available?

All the data related to the manuscript are available as supplementary materials and at the NCBI site “accession no. GSE133866 https://www.ncbi.nlm.nih.gov/geo/query/acc.cgi?acc=GSE133866”

Reviewer#1

1.In abstract: “However, despite the salinization issue the production of rice is required to be

increased from the current figure of ~500 million tons to ~800 million tons by 2050 to meet the demand of the ever growing population”. Being general statement, it should be deleted.

The concerned sentence has been modified (Please see L 27-30)

2.In Materials and Methods: Kindly cite the reference mentioning that how the authors have chosen these two

cultivar which it shows have contrast salt adaptive. Pl cite the references for Pokkali and badami as sensitive to

salinity.

References cited, as desired (Please see L521-522)

3.Line # 541: Replace the 1.5% Nacl to EC value which is more acceptable form of salinity unit

Reviewer#2 has advised to represent the NaCl concentration in mM, which has been done. Salinity as EC is generally for TDS, which is not the case here. However, as desired by the Reviewer#1, the EC value for 256 mM NaCl (1.5% NaCl) has been given in parenthesis at the first mention of the NaCl concentration (Please see L130). Reference to this has been cited in the Materials and Methods section (Please see L528-529)

4. It is wrong to treat the two halophytes at 1.5 % as in this concentration no stress will be induced in two

halophyte. Thus comparing them with rice in the same concentration will be conceptually wrong.

It is true that halophytes can tolerate high level of salinity. However, it may be noted that the two halophytes in the present case were grown without salt, and thus addition of 1.5 % NaCl, nearly 50% of the seawater salinity, is certainly going to generate response in them in order to adjust to the saline environment. And this is what is required in salt response studies, which is generally carried out at sub-lethal dose. Besides, for comparative study, it was necessary to keep the NaCl treatment concentration similar in all the plants tested. 

5.The author should cite and discuss the most related following two papers; a) Mondal TK, Ganie SA, Debnath AB(2015) Identification of novel and conserved miRNAs from extreme halophyte, Oryza coarctata, a wild relative of rice. PLoS ONE 10(10): e0140675. doi:10.1371/ journal.pone.0140675and b) Mondal TK, Panda AK, Rawal HC, Sharma T R (2018) Discovery of microRNA-target modules of African rice (Oryza glaberrima) under salinity stress. Scientific Reports 8, Article number: 570

The suggested papers have been referred and cited at suitable places (Please see the citations 39 and 55)

6.What is the need of fragmented salinity treatment described in line number 562-568?

The fragmented treatment was given to allow the plant to adjust to the changing environment and to avoid salt stress shock. This has been mentioned in the manuscript (L529-531, L553)

7.Line #654. Which actin gene was used as internal control? . Kinldy write the LOC number.

LOC (LOC_Os03g50885) number of actin has been given (L639)

8.Finally since this is the sequence based research and I could not check the sequence as it was not submitted to NCBI and open for review. This is also against the policy of PLOS one

The raw sRNA data are available at the NCBI as the accession no. GSE133866

https://www.ncbi.nlm.nih.gov/geo/query/acc.cgi?acc=GSE133866

This has been mentioned under the data availability section of the online submission process/format

Review#2

The manuscript is well written and easy to read. Large number of data are presented in 8 figures and 7 supporting files and the obtained results support the final conclusions. Unfortunately, almost all data presented are without the number of replication and without bars for standard error or deviation.

The data have been presented without replication only for the Northern (Fig. 1, 2 and 3) and the leaf sections for visualization of miRNA expression (Fig. 8). These should be quite acceptable. In other figures the data are presented as mean � standard deviation. Fig. 7 has been revised showing DAB staining in three leaves for each case.

1. Abstract and discussion sections are long. Abstract should focus on the scope of the work and obtained results rather then on “ever growing population” (row 30) or world rice production (row 29).

The concerned sentence has been changed (L29-30). The abstract is within the permissible word limit of 300. The discussion section has been shortened by 10 lines. 

a. Why authors decided to express the concentration of applied NaCl in % and not in mM as in the major number ofpublications?

NaCl concentration changed to mM.

b. Will be much easier to follow the text if data in Fig. 1 and 3 are presented in accelerating order regarding the miRNAs numbers. In the current version is difficult to find the respective miRNA on the figures. For example miR169 is following miR396 (Fig. 1) and miR12477 is after miR12478 (Fig. 3).

The miRNAs have been arranged in ascending order, as desired

c. Some of the photos on Fig.3 are with bad quality (for example miR12482) and is difficult to accept the conclusions for the relative expression level. Definitely, some of the Northern must be repeated or (if available) better photos should be presented. Again, on Fig.3 differences in the expression levels of miR12475 and miR12477 are minimal and not sufficient to claim up-regulation in O. coarctata (row 249, 250). In this case, to enforce their conclusions, authors have to specify the number of Northern experiments that were done for each miRNA.

The Northern of miR12478 and miR12482 has been repeated. The concerned sentences on the relative expression of miR12475 and miR12477 have been modified in the text (L244-246)

d. Fig. 6. What mean the control sample with relative expression of 1? The normal way of presentation of RT-qPCR results is 2-ΔCT.

Yes, the result of RT-qPCR is presented as 2-�CT. But the 2-�CT value is converted into fold change, given by 2^-ΔΔCt, and the result is expressed as fold change in expression in Treated over Control. In such case the value of control is always taken as unity. Any 2^-ΔΔCt value more than one, i.e. control, is considered as upregulation and less than one is considered as downregulation. As such there is no need of showing the value of the control, which is unity, but I have given for the sake of clarity. 

e. Fig 7. On the upper panel scale bar is missing to confirm that the stained leaves have the same size. In this case, instead of the one dimensional and very rough densitometrical analysis, some simple but precise method for quantification of H2O2 content in DAB stained tissue should be applied (for example Ramel et al., 2009). In the lower panel of Fig. 7 the bars for standard error or deviation are missing, together with the total number of stained leaves.

We have done DAB staining in a way similar to Ramel et al. 2009. We have repeated the experiment, and photographed the leaves putting a scale on one side. Results of three staining have been presented. Densitometric analysis has not been done, as it can be done for only one sample.

3. Material and methods are long and scholastic for the level of PLOS. For example:

Row 673: The PCR tube was placed in a PCR machine…

Row 724 : The entire PCR product (50 �L) was electrophoresed on 1.5 % agarose gel….

Row 741: dissolving 50 mg in 45 ml… (only the final concentration has to be reported).

Definitely, large part of this section must be revised and shortened.

Material and methods section has been revised to take care of the above comments (Please see L655-657, 697-701, 714-715). However, although the minute details in the Material and Methods section has been removed from several places, the section as such could not be shortened much (not more than 12 lines), as the Northern, dual luciferase assay and 5’RACE PCR had to be described to clarity. I hope the modification done would be acceptable

4. References. Unfortunately, all the references are presented in inappropriate way. According to the “instructions for authors” PLOS uses the reference style outlined by the International Committee of Medical Journal Editors (ICMJE). Example format:

Hou WR, Hou YL, Wu GF, Song Y, Su XL, Sun B, et al. cDNA, genomic sequence cloning and overexpression of

ribosomal protein gene L9 (rpL9) of the giant panda (Ailuropoda melanoleuca). Genet Mol Res. 2011;10: 1576-1588.

The references have been thoroughly checked and formatted as per the journal’s requirement

6. PLOS authors have the option to publish the peer review history of their article (what does this mean? ). If

published, this will include your full peer review and any attached files.

Not applicable to the authors

---

## [Decision Letter · Decision Letter 1]

13 Mar 2020

Identification and expression analysis of miRNAs and elucidation of their role in salt tolerance in rice varieties susceptible and tolerant to salinity

PONE-D-19-25289R1

Dear Dr. Shaw,

We are pleased to inform you that your manuscript has been judged scientifically suitable for publication and will be formally accepted for publication once it complies with all outstanding technical requirements.

With kind regards,

Alberto Amato

Academic Editor

PLOS ONE

Additional Editor Comments (optional):

One of the reviewers has never submitted his report, hence as academic editor I decided to rely only on the report of Reviewer 2, who was happy with the new version you submitted (as you can see below).

Reviewers' comments:

Reviewer's Responses to Questions

**Comments to the Author**

1. If the authors have adequately addressed your comments raised in a previous round of review and you feel that this manuscript is now acceptable for publication, you may indicate that here to bypass the “Comments to the Author” section, enter your conflict of interest statement in the “Confidential to Editor” section, and submit your "Accept" recommendation.

Reviewer #2: All comments have been addressed

2. Is the manuscript technically sound, and do the data support the conclusions?

Reviewer #2: Yes

3. Has the statistical analysis been performed appropriately and rigorously? 

Reviewer #2: Yes

4. Have the authors made all data underlying the findings in their manuscript fully available?

Reviewer #2: Yes

5. Is the manuscript presented in an intelligible fashion and written in standard English?

Reviewer #2: Yes

6. Review Comments to the Author

Reviewer #2: The new version of the manuscript “Identification and expression analysis of miRNAs and elucidation of their role in salt tolerance in rice varieties susceptible and tolerant to salinity” is definitely improved. Authors improved also the quality of the figures and new supplementary figures are now included, although, to my opinion, the new S4 and S6 figures (Uncropped Northern blot images of known and novel miRNAs) did not contribute for the better understanding of the manuscript. Authors replied to all my comments and made the suggested changes in the text.

7. PLOS authors have the option to publish the peer review history of their article (what does this mean?). If published, this will include your full peer review and any attached files.

Reviewer #2: No